# Characterization of Plasmonic Scattering, Luminescent Down-Shifting, and Metal-Enhanced Fluorescence and Applications on Silicon Solar Cells

**DOI:** 10.3390/nano11041013

**Published:** 2021-04-15

**Authors:** Wen-Jeng Ho, Jheng-Jie Liu, Jhih-Ciang Chen

**Affiliations:** Department of Electro-Optical Engineering, National Taipei University of Technology, No. 1, Section 3, Zhongxiao East Road, Taipei 10608, Taiwan; jjliu@mail.ntut.edu.tw (J.-J.L.); jo840112@gmail.com (J.-C.C.)

**Keywords:** Eu-doped phosphors, fluorescence emission, luminescent down-shifting, metal enhanced fluorescence, surface plasmon resonance, silver nanoparticles, silicon solar cells

## Abstract

This paper studied characterized the plasmonic effects of silver nanoparticles (Ag-NPs), the luminescent down-shifting of Eu-doped phosphor particles, and the metal-enhanced fluorescence (MEF) achieved by combining the two processes to enhance the conversion efficiency of silicon solar cells. We obtained measurements of photoluminescence (PL) and external quantum efficiency (EQE) at room temperature to determine whether the fluorescence emissions intensity of Eu-doped phosphor was enhanced or quenched by excitation induced via surface plasmon resonance (SPR). Overall, fluorescence intensity was enhanced when the fluorescence emission band was strongly coupled to the SPR band of Ag-NPs and the two particles were separated by a suitable distance. We observed a 1.125× increase in PL fluorescence intensity at a wavelength of 514 nm and a 7.05% improvement in EQE (from 57.96% to 62.05%) attributable to MEF effects. The combined effects led to a 26.02% increase in conversion efficiency (from 10.23% to 12.89%) in the cell with spacer/NPs/SOG-phosphors and a 22.09% increase (from 10.23% to 12.48%) in the cell with spacer/SOG-phosphors, compared to the bare solar cell. This corresponds to an impressive 0.85% increase in absolute efficiency (from 12.04% to 12.89%), compared to the cell with only spacer/SOG.

## 1. Introduction

Most efforts to further the development of silicon-based solar cells have focused on enhancing conversion efficiency and reducing overall costs [1,2,3,4]. The conversion efficiency of crystalline silicon solar cells can be improved via light trapping using pyramidal surface structures or anti-reflective coatings [5,6]. A number of studies have also demonstrated that enhanced light trapping effects that can be achieved via localized surface plasmon resonance are induced by metallic nanoparticles (NPs), such as gold (Au-NPs), silver (Ag-NPs), and aluminum (Al-NPs), applied to the front or rear surface [7,8,9,10,11,12,13].

However, note that the theoretical maximum conversion efficiency of single-junction crystalline silicon solar cells is 31% [14,15] due to the bandgap energy (E_g_) of 1.1 eV. Photons with energy below E_g_ are not absorbed (transmission loss) and photons with energy exceeding E_g_ are absorbed; however, much of the excess energy is lost via thermalization loss by generated carriers [16,17]. Materials must be selected carefully to minimize mismatch between the solar spectrum and the spectral absorption of solar cell (spectral losses). One approach to expand the usable range of light involves the application of a conversion layer, such as down conversion (DC) [18,19], down shifting (DS) [20,21], up conversion (UC) [22,23], and multi-junction tandem structures [24,25]. DC involves converting an incident high-energy photon (UV-blue wavelengths) into two or more photons of lower energy (within visible wavelengths). DS is similar to DC, but only a single photon is emitted. UC involves converting two or more photons of low energy (NIR-IR wavelengths) into a single high-energy photon (within visible wavelengths). Note that the conversion efficiency of crystalline silicon solar cells at UV-blue wavelengths is relatively low due to high reflectance and carrier recombination at the surface as well as the low responsivity of silicon semiconductors. Applying a dielectric layer containing DC or DS luminescent materials on the front-side surface of silicon solar cells can reduce spectral losses, reflectance, and surface recombination, while enhancing the conversion efficiency. A small number of studies have examined the plasmonic scattering of Ag-NPs and the luminescent down shifting (LDS) of Eu-doped phosphors to enhance the conversion efficiency of crystalline silicon solar cells [26,27]. The integration of Ag and Eu-doped phosphors on silicon solar cells has also been shown to enhance conversion efficiency via metal-enhanced fluorescence (MEF) [28,29,30].

This paper systematically characterizes the effects of Ag-NPs, Eu-doped phosphor particles, and their combination in terms of absorbance, photoluminescence (PL), and optical reflectance spectra. These effects were also shown to enhance the efficiency of silicon solar cells, as confirmed by external quantum efficiency (EQE) response and photovoltaic current density–voltage (J-V) measurements under one-sun AM 1.5 G. MEF effects in this study achieved a 26.02% increase in efficiency and 0.85% increase in absolute efficiency.

## 2. Experimental Methods

### 2.1. Preparation and Characterization of Plasmonic Silver Nanoparticles

We first characterized the plasmonic effects induced by Ag-NPs of various diameters by preparing two groups of test samples. Group 1: A 20-nm-thick SiO_2_ spacer layer was deposited on six clean quartz and silicon substrates. Films of Ag were deposited to a thickness of 3, 5, and 7 nm over the spacer layer using E-beam evaporation. Then, the samples were annealed at 200 °C for 30 min under ambient H_2_ to form Ag-NPs of various diameters on the spacer layer. Group 2: Three duplicate samples from Group 1 were coated (over the Ag-NP layer) with a SiO_2_ capping layer to a thickness of 74 nm using e-beam evaporation. The diameter and surface coverage of Ag-NPs of all samples in Group 1 were examined via scanning electron microscopy (SEM; JEOL JSM-6500F, JEOL Ltd., Tokyo, Japan) in conjunction with Image-J software. We also examined the plasmonic effects (localized surface plasmon resonance (LSPR)) produced by the Ag-NPs by measuring the absorbance spectra of test samples on a quartz substrate using a spectrometer (USB 4000, Ocean Optics, Inc., Largo, FL, USA) as well as the reflectance spectra using a UV/VIS/NIR spectrophotometer (PerkinElmer LAMBDA 35, Waltham, MA, USA).

The degree to which the plasmonic effects affected photovoltaic performance was assessed by coating planar silicon solar cells with Ag-NPs of various diameters. Bare planar solar cells were fabricated using the methods reported in a previous study [26,27]. A SiO_2_ spacer layer was deposited to a thickness of 20 nm on the front surface of the bare solar cells. Ag films were deposited over the spacer layer to a thickness of 3, 5, or 7 nm and then subjected to annealing at 200 °C for 30 min under ambient H_2_ to form Ag-NPs of various diameters. The NP layer was coated with a SiO_2_ capping layer to a thickness of 74 nm using e-beam evaporation. The total thickness of the coatings on the substrate (spacer and capping layers) was 94 nm. Note that the Ag-NPs embedded in the coatings formed a plasmonic anti-reflection coating (ARC) for the silicon solar cells. Figure 1a presents a schematic diagram of silicon solar cells with Ag-NPs on the surface or embedded in the SiO_2_ layer. The optical reflectance, EQE, and photovoltaic J-V of the cells were measured at each processing stage to confirm the occurrence of plasmonic effects and their influence on solar cell performance.

### 2.2. Preparation and Characterization of Eu-Doped Phosphor Species

Next, we characterized the fluorescence performance of Eu-doped phosphors by coating clean silicon substrates with a silicate solution of SiO_2_ (SOG; Emulsitone Company product, Whippany, NJ, USA) containing Eu-doped phosphor species (InteMatix Company product, Fremont, CA, USA) at a concentration of 3 wt %. Note that the coating solution was prepared by mixing 1.94 g of silicate solution with 0.06 g of various species of Eu-doped phosphors (species-500, species-550, or species-600). The resulting SOG:Eu-doped phosphor solutions were spin-coated onto clean silicon substrates at 3000 rpm for 60 s and then baked at 200 °C for 30 min under a clean dry air atmosphere. The average diameter of the phosphor particles was 15 μm, and the coverage on the test samples was 8% [31]. We examined the fluorescence emission of samples with various species of Eu-doped phosphor by measuring the PL (Ramboss 500i Micro-PL spectroscopy, DONGWOO Optron, Gwangju-si, Korea) at room temperature. The excitation light source used in the PL system was a solid-state UV laser with emission wavelength of 266 nm and emission power of 20 mW. The excitation band of the Eu-doped phosphor species was roughly 200 to 450 nm [32].

The degree to which the luminescent down shifting affected photovoltaic performance was assessed by spin-coating planar silicon solar cells with an SOG layer containing Eu-doped phosphors (SOG:Eu-doped phosphor). Note that a SiO_2_ spacer layer was first deposited to a thickness of 20 nm on the front surface of bare silicon solar cells using E-beam evaporation. Then, SOG solutions containing Eu-doped phosphors (species-500, species-550, and species-600) were spin-coated to a thickness of 200 nm over the SiO_2_ spacer layer. Then, the cells were baked at 200 °C for 30 min under a clean dry air atmosphere to remove organic solvents. Figure 1b presents a schematic diagram of a silicon solar cell with an SOG:Eu-doped phosphor layer. The optical reflectance, EQE, and photovoltaic J-V of the cells were measured at each processing stage to confirm the occurrence of LDS and characterize its effects on solar cell performance.

### 2.3. Sample Preparation and Characterization of Metal-Enhanced Fluorescence Emissions

We also characterized the effects of Eu-doped phosphor incorporated with Ag-NPs in terms of enhanced or quenched fluorescence emissions. A 20-nm-thick SiO_2_ spacer layer was deposited on silicon substrates, over which we deposited a layer of Ag to a thickness of 3, 5, or 7 nm via E-beam evaporation. Then, the samples were annealed at 200 °C for 30 min under ambient H_2_ to form Ag-NPs. Finally, SOG solution diluted with ethanol at a ratio of 2:8 was mixed with Eu-doped phosphors (species-500, species-550, or species-600) powder (3 wt %) and spin-coated over the Ag-NP layer at 3000 rpm for 60 s and baked at 200 °C for 30 min under an air atmosphere to remove organic solvent.

Figure 1c presents a schematic diagram of a silicon solar cell with a metal-enhanced fluorescence layer. The optical reflectance, EQE, EQE enhancement (Δ*EQE*), and photovoltaic J-V of the cells at each processing stages were measured to confirm the effects of metal-enhanced fluorescence emissions on solar cell performance. Note that all reported data were averaged from three measurements. Overall, the diameter of the Ag-NPs was proportional to the thickness of the Ag films, and the distance between the Ag-NPs and Eu-doped phosphor particles was shown to affect (enhance or quench) the fluorescence emission characteristics, as estimated using PL measurements at room temperature. Our results guided the formulation of a model describing metal-enhanced fluorescence emissions.

## 3. Results and Discussion

### 3.1. Effects of Plasmonic Silver Nanoparticles on Conversion Performance

The LSPR induced by metallic nano-size particles provided impressive near-field light concentration and far-field scattering. The plasmonic resonance of NPs can be modified by adjusting the size, shape, constituent materials, and surrounding medium. Figure 2 presents SEM images and the size distribution of Ag-NPs in Ag films deposited to thicknesses of 3, 5, or 7 nm. The average diameter and coverage of Ag-NPs were as follows: Ag: 3 nm (18.21 nm and 38.43%), Ag: 5 nm (21.91 nm and 45.93%), and Ag: 7 nm (32.59 nm and 49.77%). The average diameter and coverage of the Ag-NPs were proportional to the thickness of the Ag films. The size distribution of the Ag-NPs ranged from 5 to 60 nm.

Excitation by light at a specific wavelength was shown to induce the collective oscillation of conduction electrons on Ag-NP surfaces, in a phenomenon known as LSPR. Figure 3 presents the absorbance spectra of samples with the following configurations: (a) Ag-NPs/SiO_2_ spacer/quartz-substrate and (b) SiO_2_-cap/Ag-NPs/SiO_2_-spacer/quartz-substrate. The broad absorption range (350 to 500 nm) can be attributed to the broad range of NP diameters (5 to 60 nm). Note that NPs of larger diameter produced peaks of higher intensity with a distinct red-shift in the peak wavelength. The absorption range and intensity were further enhanced by the SiO_2_ capping layer, which altered the refractive index of the medium surrounding the Ag-NPs from 1.0 (air) to 1.45 (SiO_2_).

Figure 4 presents the optical reflectance of solar cell devices with the following configurations: bare silicon solar cell, bare cell with a 20-nm-thick SiO_2_ spacer layer, cells with Ag-NPs of various diameter on a SiO_2_ spacer layer, and cells capped with an NP layer and SiO_2_ capping layer. For the sake of comparison, we also included a cell coated with a 94-nm-thick SiO_2_ spacer + capping layer (i.e., without NPs). Table 1 lists the average weighted reflectance (*R_w_*) of all cells calculated at wavelengths from 350 to 1000 nm. The reflectance of the cell coated with a SiO_2_ spacer layer was lower than that of the bare cell due to the anti-reflective property of the SiO_2_. The plasmonic effects of the Ag-NPs further reduced reflectance to below that of the cells with only a spacer layer. Note that the sharp drop in reflectance (at wavelengths of 350 to 500 nm) can be attributed to the SPR absorption of Ag-NPs. The drop in reflectance (at wavelengths of 550 to 1000 nm) can be attributed to the plasmonic forward scattering of incident photons by Ag-NPs. Our results revealed that samples with Ag-NPs of larger diameter presented lower reflectance due to an increase in SPR absorption. Increasing the diameter of the Ag-NPs was also shown to red-shift the trough in the reflectance spectrum, which is in good agreement with the absorbance spectra.

Figure 5 presents the EQE response of a bare silicon solar cell, a cell with a spacer layer, cells with spacer/NPs (various diameters), and cells with spacer/NPs/capping layer (SiO_2_). For the sake of comparison, we also included a cell coated with a 94-nm-thick sample comprising a SiO_2_ spacer + capping layer (i.e., without NPs). Table 1 lists the average weighted EQE (*EQE_w_*) of all cells calculated at wavelengths from 350 to 1100 nm. At wavelengths of 300 to 375 nm, the EQE values of the cell with only a spacer layer and the cell with a spacer/NPs exceeded those of the bare cell, due to the passivation effects of SiO_2_ and the lower reflectance of SiO_2_ capping layer. Cells with NPs without the capping layer presented EQE values slightly lower than those of the bare cell at wavelengths from 375 to 425 nm, due to the SPR absorption of the NPs. However, note that the EQE values were higher for incident photons with a wavelength exceeding 450 nm, due to an increase in the optical length of incident photons in the silicon active layer resulting from plasmonic forward scattering induced by Ag-NPs. The EQE values of cells with a spacer/NPs/capping layer were even higher due to the anti-reflection effects of the SiO_2_ capping layer. The EQE response values are in good agreement with the optical reflectance results. Taken together, it appears that the EQE response of solar cells can be enhanced by plasmonic effects induced by Ag-NPs of various diameters.

Figure 6 presents the photovoltaic J-V curves of a cell with a spacer layer, cells with spacer/NPs (various diameters), and cells with spacer/NPs/capping layer (SiO_2_). Table 1 summarizes the photovoltaic performance of all solar cells evaluated in this study. The open-circuit voltage (V_oc_) values of the cell with only the spacer and cells with spacer/NPs were 3 to 11 mV higher than that of the bare solar cell due to the passivation effects of SiO2 on the Si surface. The short-circuit current–density (J_sc_) and conversion efficiency (η) values were as follows: bare solar cell (26.26 mA/cm^2^, 10.23%), spacer only (27.69 mA/cm^2^, 10.84%), spacer/NPs (diameter: 3, 5, and 7 nm) (28.56 mA/cm^2^/11.07%, 29.18 mA/cm^2^/11.28%, 29.81 mA/cm^2^/11.55%), and spacer/NPs/capping layer (NP diameter: 3, 5, and 7 nm) (31.42 mA/cm^2^/12.17%, 31.73 mA/cm^2^/12.32%, 31.96 mA/cm^2^/12.38%). The J_sc_ values corresponded to the EQE response values. The J_sc_ and η values of cells with spacer/NPs were higher than those of the cell with only a spacer layer, due to the plasmonic effects of the Ag-NPs. The J_sc_ and η values of cells with spacer/NPs/capping layer (31.42–31.96 mA/cm^2^, 12.17% to 12.38%) were also higher than those of the cell with only a SiO_2_ ARC (31.35 mA/cm^2^, 12.04%), which was due to the plasmonic effects of the Ag-NPs embedded in SiO_2_. The insets in Figure 6 show that the enhanced J_sc_ was dependent on the plasmonic effect and diameter of Ag-NPs. Our results demonstrate that applying Ag-NPs (Ag: 7 nm) to induce plasmonic effects improves the conversion efficiency by 21.02% (from 10.23% to 12.38%) compared to bare solar cells.

### 3.2. Effects of Eu-Doped Phosphor Species on Conversion Performance

Figure 7 presents the PL spectra of Eu-doped phosphors (species-500, species-550, and species-600) and the responsivity of silicon p-n junctions under excitation by a light source with a wavelength of 266 nm. The PL emission peak wavelength and full width at half-maximum (FWHM) of the Eu-doped phosphor species were as follows: species-500 (514 nm, 62 nm), species-550 (544 nm, 90 nm), and species-600 (599 nm, 70 nm). The absorption band of the Eu-doped phosphors in this work ranged from 200 to 450 nm. Under these conditions, the Eu-doped phosphors absorbed the incident photons (266 nm) and re-emitted photons at visible wavelengths (i.e., LDS). The absorption of down-shifted photons (re-emitted) were absorbed closer to the active depletion region of the solar cell, which increased the collection of photo-generated electron–hole pairs and suppressed the recombination of charge carriers near the surface region. This explains how the Eu-doped phosphor layer enhanced the J_SC_ and η of the silicon solar cells. The wavelength of photons emitted from species-600 exceeded the wavelengths from species-550 and species-500. The responsivity of the silicon p-n diodes was approximately 0.15, 0.20, and 0.26 A/W at the peak emission wavelength of species-500, species-550, and species-600, respectively. Therefore, it is reasonable to expect that the EQE and η of solar cells containing phosphors of a longer emission-wavelength (species-600) would exceed those coated with phosphors of a shorter emission-wavelength (i.e., species-550 and species-500).

Figure 8 presents the reflectance spectra of a bare silicon solar cell, a cell with spacer/SOG, and cells with spacer/SOG-phosphors (species-500, species-550, and species-600). Table 2 lists the *R_w_* of all cells calculated at wavelengths from 350 to 1100 nm. The reflectance of a solar cell with spacer/SOG displayed antireflective characteristics, with the lowest reflectance at a wavelength of approximately 575 nm, due to destructive interference. The reflectance of the solar cells spacer/SOG-phosphors was lower than that of the bare solar cell across the entire range of wavelengths. The reflectance of cells spacer/SOG-phosphors was lower than that of the cell with spacer/SOG across wavelength ranges of 350 to 500 nm and 650 to 1000 nm. This can be attributed to the absorption of high energy photons (350 to 500 nm) by the Eu-doped phosphor particles, due to LDS effects and the forward scattering of low energy incident photons (at wavelengths of 650 to 1000 nm). Across a wavelength range of 515 to 650 nm, the reflectance of the cells with spacer/SOG-phosphors was higher than that of the cell with a spacer/SOG, due to weaker destructive interference. We expected that LDS effects and forward light scattering would further enhance the EQE response and photovoltaic performance of solar cells.

Figure 9 presents the EQE spectra of a bare silicon solar cell, a cell with spacer/SOG, and cells with spacer/SOG-phosphors (species-500, species-550, and species-600). The EQE of cells with spacer/SOG was higher than that of the bare cell at wavelengths of 350 to 1100 nm, which was due to the anti-reflection effects of the SiO_2_. The EQE responses of cells with spacer/SOG-phosphors exceeded that of the cell without phosphors. The EQE responses of all of the cells corresponded to the observed changes in optical reflectance. The EQE response of the cell with longer emission-wavelength phosphors (species-600) slightly exceeded that of the cells with shorter emission-wavelength phosphors (species-550 > species-500), which is in agreement with the responsivity of silicon as a function of incident photon wavelength. Table 2 lists the *EQE_w_* of all cells calculated at wavelengths from 350 to 1100 nm. These results indicate that the spacer/SOG-phosphors enhanced the EQE response of the solar cells while providing a corresponding increase in photovoltaic performance.

Figure 10 presents the photovoltaic J-V curves of a bare silicon solar cell, a cell spacer/SOG, and cells with spacer/SOG-phosphors (species-500, species-550, and species-600). Table 2 summarizes the photovoltaic performance of all solar cells evaluated in this study. The J_sc_ values of the cells with spacer/SOG-phosphors exceeded those of the cell with spacer/SOG and the bare cell, due to LDS effects and forward light scattering. As shown in the inset of Figure 10, the J_sc_ values increased following the inclusion of phosphor particles in the SOG layer as a function of phosphor species. The J_sc_ of the cell with longer emission-wavelength phosphors (species-600) exceeded that of cells with shorter emission-wavelength phosphors (species-550 > species-500), which is in good agreement with EQE response. Overall, η was proportional to J_sc_. The highest η (12.54%) and J_sc_ (32.32 mA/cm^2^) values were obtained from the cell with species-600 phosphors, exceeding those of the cell with SOG (12.04%, 31.35 mA/cm^2^) by 0.5% and the bare cell (10.23%, 26.26 mA/cm^2^) by 2.3%. These results demonstrate that the inclusion of phosphors (species-600) increased conversion efficiency by 23.08% (from 10.23% to 12.54%), compared to the bare cell.

### 3.3. Effects of Metal-Enhanced Fluorescence on Conversion Performance

Figure 11a presents a schematic diagram showing Ag-NPs and Eu-doped phosphor particles deposited on the front surface of a silicon substrate. The diameter of the Ag-NPs depended on the thickness of the initial Ag film prior to thermal annealing. Note that the Ag-NPs were deposited on a SiO_2_ spacer layer. Thus, if the plane of the SiO_2_ spacer layer was horizontal, then the height of larger Ag-NPs would exceed that of smaller NPs. Under these conditions, the distance between the NPs and phosphor in the SOG layer varied as a function of NP size, as follows: d_1_ (Ag: 3 nm) > d_2_ (Ag: 5 nm) > d_3_ (Ag: 7 nm). Note that an excessively small distance between particles in the two layers would cause the transmission of fluorescence energy loss from Eu-doped phosphor particles to Ag-NPs. Figure 11b illustrates the proposed model, which describes metal-enhanced fluorescence (MEF) using two examples. In Case 1, fluorescence emissions are first generated by exposing Eu-doped phosphor particles to AM 1.5G illumination. The subsequent emission of photons of a specific wavelength from the Eu-doped phosphor particles induces SPR effects on the Ag-NPs. As long as Ag-NPs and phosphor particles are separated by a suitable distance, excitation from adjacent Ag-NPs will enhance the fluorescence emissions of Eu-doped phosphor particles. If the distance between the Ag-NPs and phosphor particles is too small, then energy loss from Eu-doped phosphor particles to Ag-NPs would reduce (or entirely quench) the fluorescence intensity of the Eu-doped phosphor particles. In Case 2, SPR generated in Ag-NPs exposed to AM 1.5G enhances the fluorescence emissions of Eu-doped phosphor particles separated by a suitable distance.

Figure 12 presents PL spectra from spacer/SOG-phosphor samples with and without Ag-NPs at room temperature. The peak PL emission wavelengths of samples with phosphors were centered at 514 nm (species-500), 544 nm (species-550), and 599 nm (species-600). As shown in Figure 3, the SPR wavelength range of Ag-NP samples was roughly 350 to 550 nm. In samples with an optimal distance between the fluorescent particles and the metallic particles, the range of wavelengths associated with the coupling of fluorescence emission and SPR affected the characteristic fluorescence emissions [33]. The fact that the PL emission range from species-500 phosphors was within the SPR wavelength range of the Ag: 3-nm NPs resulted in good coupling, which was referred to as MEF. As shown in Figure 12, this led to a 12.47% increase in fluorescence emissions, compared to spacer/SOG-phosphor samples (i.e., without Ag-NPs). Note that the fluorescence emission wavelengths of species-550 and species-600 phosphors fell almost entirely outside the SPR wavelength range of the NPs, resulting in weak coupling. Compared to spacer/SOG-phosphor samples, this decreased the fluorescence intensity, as follows: species-550 (−3.18%) and species-600 (−10.58%). Our results revealed that species-550 and species-600 phosphors were unable to generate MEF effects, regardless of the size of Ag-NPs. Furthermore, the fluorescence intensity of the sample with 7-nm Ag-NPs was less than that of the sample with 5-nm Ag-NPs. Thus, subsequent analysis on photovoltaic performance focused on the combination of samples with phosphors (species-500) and Ag-NPs of various sizes (3, 5, and 7 nm).

Figure 13 presents the optical reflectance spectra of a bare silicon solar cell, a cell with spacer/SOG, a cell with spacer/SOG-phosphors, and cells with spacer/NPs/SOG-phosphors. The reflectance of the cells with only SOG was the lowest at a wavelength of approximately 575 nm, which was due to destructive interference. The reflectance of the cell with spacer/SOG-phosphors (i.e., without Ag-NPs) was lower than that of the bare cell across the entire range of wavelengths. Reflectance was particularly low over wavelength ranges of 350 to 500 nm (due to LDS effects) and 650 to 1000 nm (due to forward light scattering). The reflectance of cells with NPs presented the lowest overall reflectance across a wavelength range of 400 to 650 nm. The cell with larger NPs provided the most pronounced reduction in reflectance. Table 3 lists the *R_w_* of all cells calculated at wavelengths from 350 to 1000 nm. The *R_w_* values of cells with spacer/NPs/SOG-phosphors were lower than those of the cells with spacer/SOG-phosphors and spacer/SOG only. Note that this difference was particularly pronounced in samples with NPs of larger diameter.

Figure 1a presents the EQE responses of a bare silicon solar cell, a cell with spacer/SOG, a cell with spacer/SOG-phosphors, and cells with spacer/NPs/SOG-phosphors. Table 3 lists the *EQE_w_* of all cells calculated at wavelengths from 350 to 1000 nm. The EQE of the cell with spacer/SOG was higher than that of the bare cell across the entire range of wavelengths, due to the antireflection effects of SiO_2_. The *EQE_w_* of the cell with spacer/NPs (3 nm) was higher than that of the cell with spacer/SOG-phosphors followed by the cell with spacer/SOG. Figure 14b presents EQE enhancement (Δ*EQE*) spectra of cells with spacer/SOG-phosphors vs. the bare cell (Δ*EQE*_1_) as well as cells with spacer/NPs/SOG-phosphors vs. the cell with spacer/SOG-phosphors (Δ*EQE*_2_). The positive values for Δ*EQE*_1_ and Δ*EQE*_2_ over a wavelength range of 400 to 520 nm can be attributed to MEF effects, whereas negative values can be attributed to the quenching of fluorescence emissions. At wavelengths beyond 600 nm, the high Δ*EQE*_1_ and Δ*EQE*_2_ values can be attributed to the scattering of incident light by Ag-NPs and phosphor particles.

Figure 15 presents the photovoltaic J-V characteristics of a bare silicon solar cell, a cell with spacer/SOG, a cell with spacer/SOG-phosphors (species-500), and cells with spacer/NPs/SOG-phosphors (species-500; NPs (Ag: 3, 5, and 7 nm)). The inset of Figure 15 shows an enlargement of J_sc_ of all evaluated cells illustrating LDS and MEF effects. The J_sc_ and η of cells with spacer/SOG (31.35 mA/cm^2^, 12.04%) and spacer/SOG-phosphors (32.22 mA/cm^2^, 12.48%) were higher than those of the bare cell (26.26 mA/cm^2^, 10.23%), which was due to the anti-reflection effects of SOG and the combined LDS and light scattering effects of phosphor particles. The J_sc_ and η of cells with spacer/NPs/SOG-phosphors (NP: 3 nm) (32.91 mA/cm^2^, 12.48%) were far higher than those of the cell with spacer/SOG-phosphors (without NPs), which was due to the MEF effects of phosphors induced by the SPR effects of Ag-NPs. Table 3 summarizes the photovoltaic performance of all evaluated solar cells. These results indicate that the J_sc_ and η values are in good agreement with the *EQE_W_* values, which was due to the fact that J_sc_ and η values are generally proportional to the EQE response of a solar cell. Note that the J_s_c and η of cells with phosphors and larger NPs (5 and 7 nm) were less than those of similar cells with phosphors and smaller NPs (3 nm). This can be attributed to the quenching of fluorescence emissions due to the loss of fluorescence energy when the distance between phosphor particles and NPs was too small. The combined effects led to a 26.02% increase in η (from 10.23% to 12.89%) in the cell with spacer/NPs/SOG-phosphors and a 22.09% increase (from 10.23% to 12.48%) in the cell with spacer/SOG-phosphors, compared to the bare solar cell. This corresponds to an impressive 0.85% increase in absolute efficiency (from 12.04% to 12.89%), compared to the cell with only spacer/SOG.

## 4. Conclusions

This study systematically analyzed the plasmonic effects of Ag-NPs, the LDS effects of Eu-doped phosphor particles, and the MEF effects obtained when the two are combined. We also examined the degree to which the various layers affect photovoltaic performance. Finally, we established an MEF model of Ag-NPs and Eu-doped phosphor particles based on measurements of PL spectra and EQE responses. A combination of Ag-NPs (Ag: 3 nm) and Eu-doped phosphor particles (species-500) produced the most pronounced MEF effects with a corresponding increase in photovoltaic performance. Compared to bare cells, we observed significant improvements in the conversion efficiency of silicon solar cells: plasmonic effects from Ag-NPs (21.02%; from 10.23% to 12.38%), LDS effects of SOG-phosphors (23.08%; from 10.23% to 12.54%), and MEF effects obtained by combining NPs with SOG-phosphors (26.02%; from 10.23% to 12.89%). This corresponds to an impressive 0.85% increase in absolute efficiency (from 12.04% to 12.89%) compared to the cell with only spacer/SOG.

## Figures and Tables

**Figure 1 nanomaterials-11-01013-f001:**
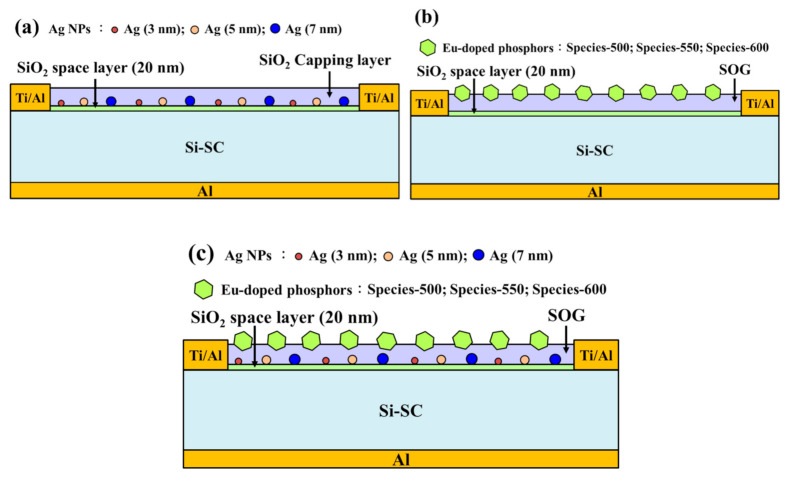
Schematic diagrams showing cells with (**a**) plasmonic Ag-NP layer, (**b**) SOG:Eu-doped phosphor layer, and (**c**) metal-enhanced fluorescence layer of Ag-NPs/SOG:Eu-doped phosphors.

**Figure 2 nanomaterials-11-01013-f002:**
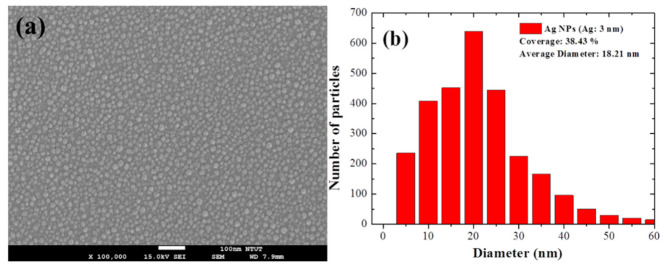
(**a**,**c**,**e**) Top view SEM images and (**b**,**d**,**f**) size distributions of Ag-NPs in Ag films deposited to thicknesses of 3, 5, and 7 nm.

**Figure 3 nanomaterials-11-01013-f003:**
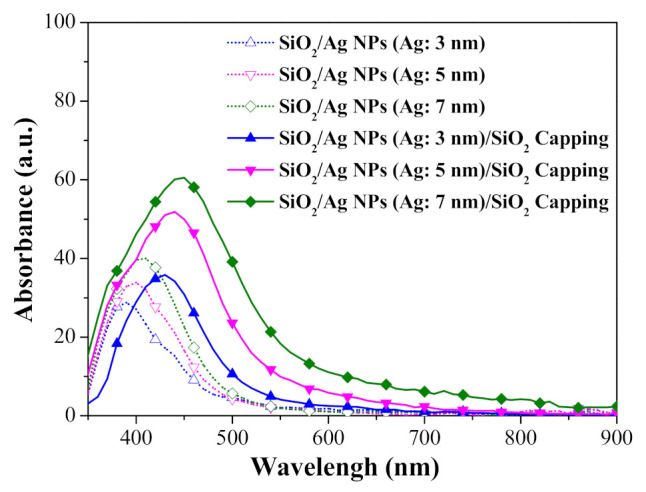
Absorbance spectra of samples with quartz-substrate/SiO_2_-spacer/Ag-NPs/SiO_2_-capping layer.

**Figure 4 nanomaterials-11-01013-f004:**
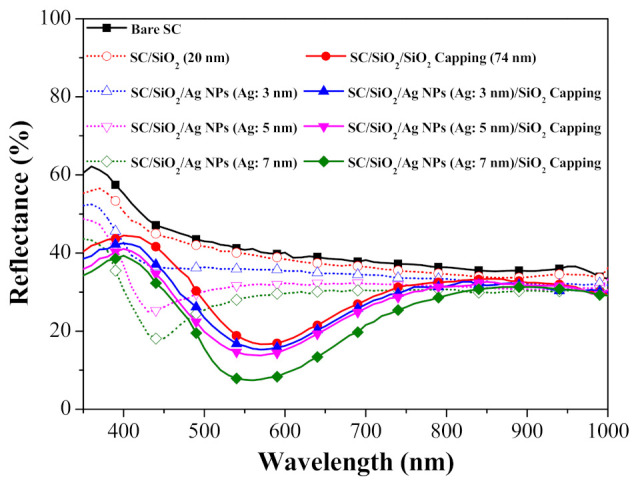
Optical reflectance of all evaluated solar cells.

**Figure 5 nanomaterials-11-01013-f005:**
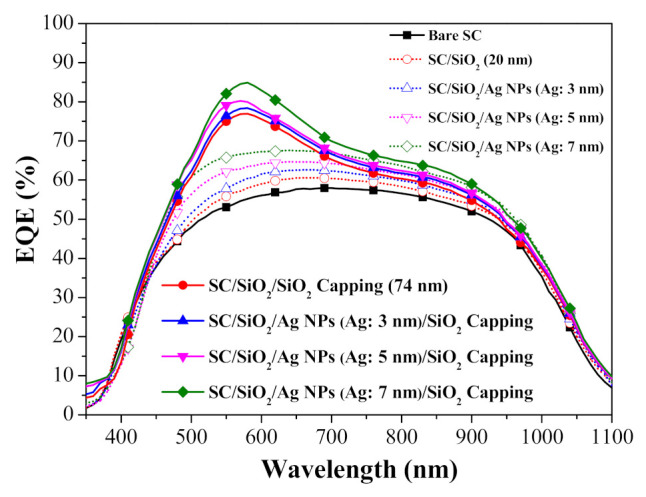
EQE responses of all evaluated solar cells.

**Figure 6 nanomaterials-11-01013-f006:**
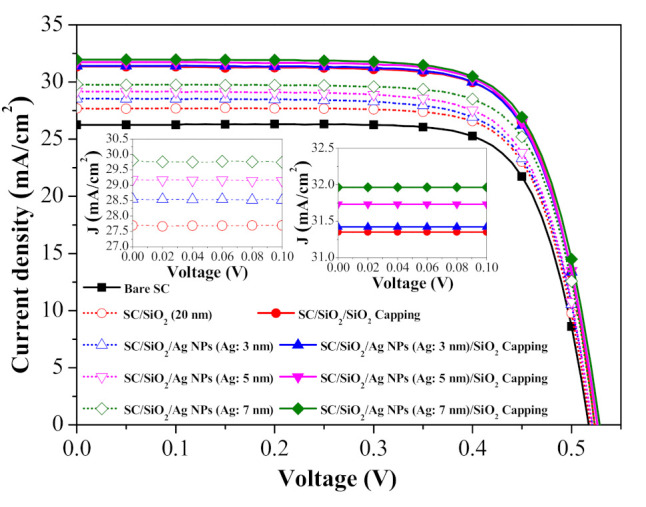
Photovoltaic J-V curves of all evaluated solar cells.

**Figure 7 nanomaterials-11-01013-f007:**
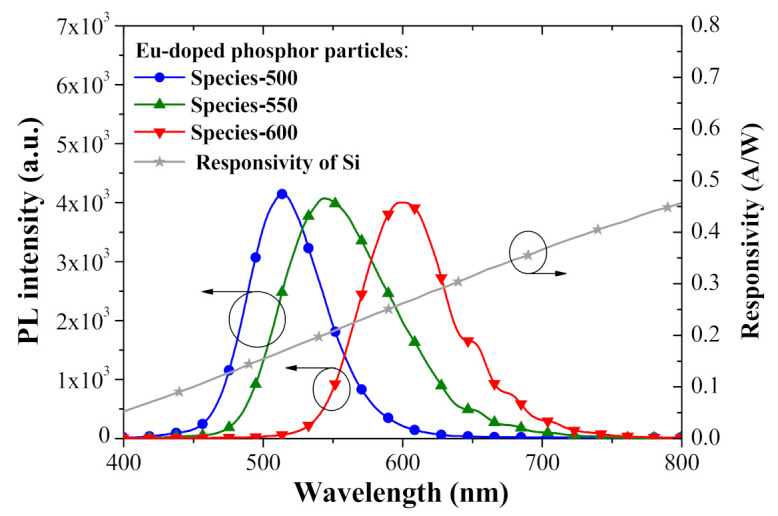
PL emission spectra of Eu-doped phosphors (species-500, species-550, and species-600) and corresponding responsivity of silicon p-n junction.

**Figure 8 nanomaterials-11-01013-f008:**
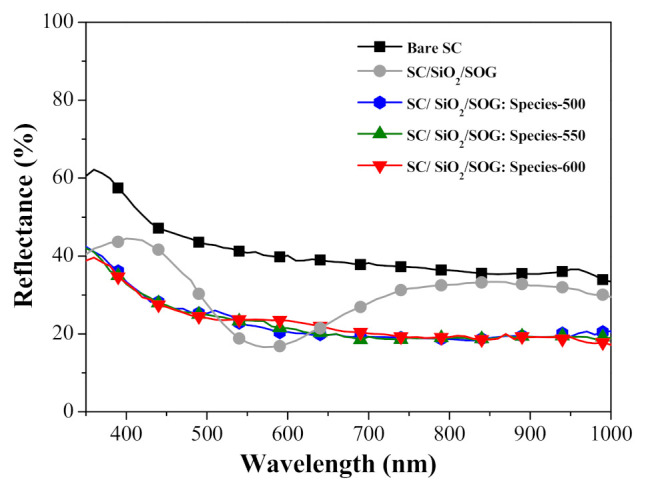
Optical reflectance spectra of bare silicon solar cell, cell spacer/SOG, and cells with spacer/SOG-phosphors (species-500, species-550, and species-600).

**Figure 9 nanomaterials-11-01013-f009:**
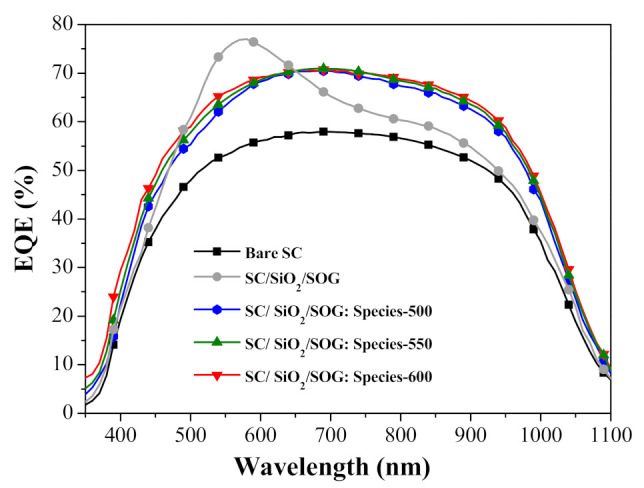
EQE spectra of bare silicon solar cell, cell with spacer/SOG, and cells with spacer/SOG-phosphors (species-500, species-550, and species-600).

**Figure 10 nanomaterials-11-01013-f010:**
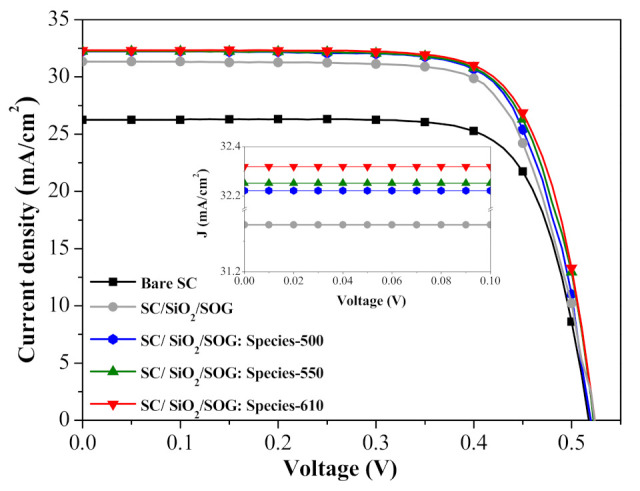
Photovoltaic J-V curves of bare silicon solar cell, cell with spacer/SOG, and cells with spacer/SOG-phosphors (species-500, species-550, and species-600).

**Figure 11 nanomaterials-11-01013-f011:**
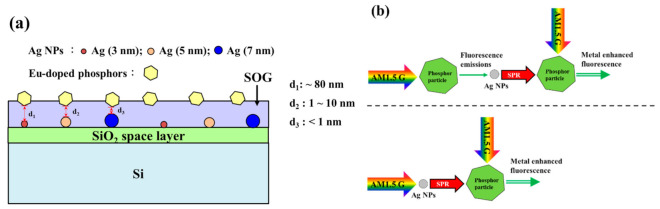
(**a**) Schematic diagram of Ag-NP and Eu-doped phosphor deposition on front surface of silicon substrate; (**b**) model of metal-enhanced fluorescence (MEF).

**Figure 12 nanomaterials-11-01013-f012:**
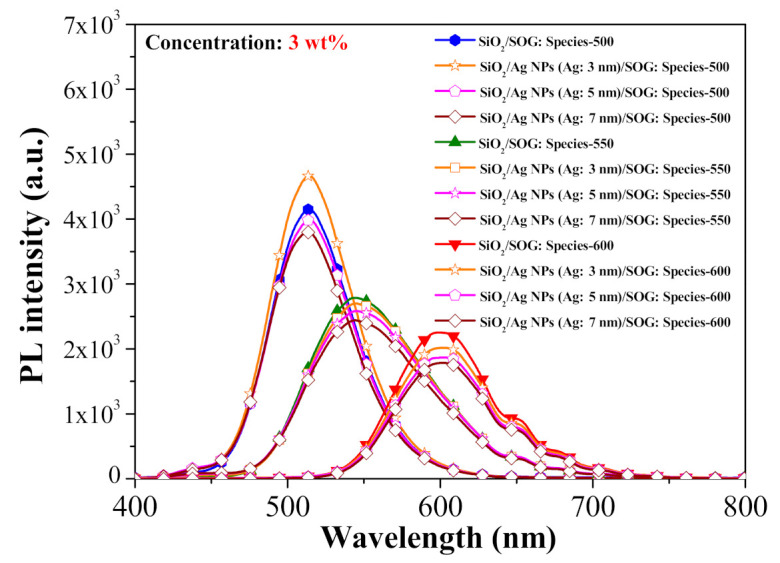
PL spectra of all samples at room temperature.

**Figure 13 nanomaterials-11-01013-f013:**
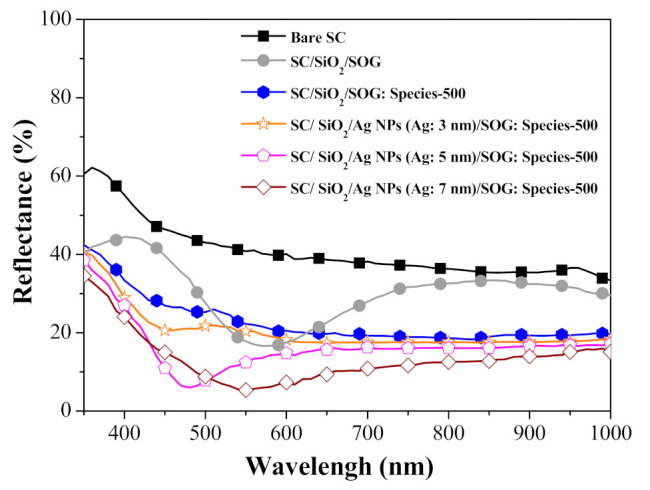
Optical reflectance spectra of bare silicon solar cell, cell with spacer/SOG, cell with spacer/SOG-phosphors (species-500), and cells with spacer/Ag-NPs/SOG-phosphors (species-500).

**Figure 14 nanomaterials-11-01013-f014:**
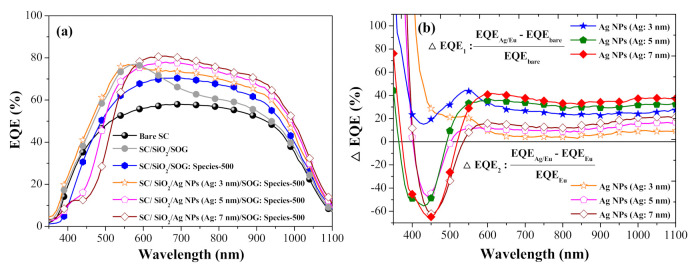
(**a**) EQE responses of bare silicon solar cell, cell with spacer/SOG, cell with spacer/SOG: phosphors (species-500), and cells with spacer/NPs/SOG: phosphors (NP: 3, 5, 7 nm); (**b**) EQE enhancement (Δ*EQE*) spectra.

**Figure 15 nanomaterials-11-01013-f015:**
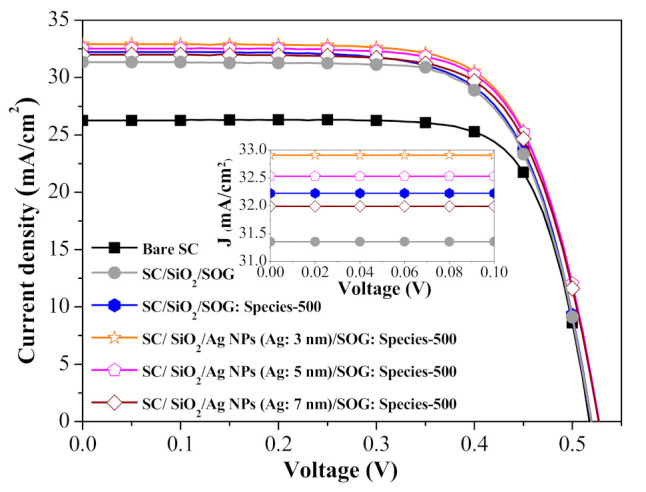
Photovoltaic J-V characteristics of bare silicon solar cell, cell with spacer/SOG, cell with spacer/SOG-phosphors (species-500), and cells with spacer/Ag-NPs/SOG-phosphors (species-500).

**Table 1 nanomaterials-11-01013-t001:** Average weighted reflectance (*R*_W_), average weighted EQE (*EQE*_W_), and photovoltaic performance of all evaluated solar cells.

Samples	R_W_ (%)	EQE_W_ (%)	J_SC_ (mA/cm^2^)	V_OC_ (mV)	FF (%)	η (%)	ΔJ_SC_ (%)	Δη (%)
	@350–1000 nm							
Bare SC (SC)	38.69	50.66	26.26	517.66	75.27	10.23	--	
SC/SiO_2_	36.82	52.24	27.69	520.75	75.15	10.84	5.45	5.96
SC/SiO_2_/Ag-NPs (Ag: 3 nm)	33.01	53.38	28.56	520.51	74.48	11.07	8.76	8.21
SC/SiO_2_/Ag-NPs (Ag: 5 nm)	29.26	56.09	29.18	520.73	74.24	11.28	11.12	10.26
SC/SiO_2_/Ag-NPs (Ag: 7 nm)	26.18	57.95	29.81	522.04	74.21	11.55	13.52	12.90
SC/SiO_2_/SiO_2_ Capping	29.59	57.96	31.35	524.92	73.15	12.04	19.38	17.69
SC/SiO_2_/Ag-NPs (Ag: 3 nm)/SiO_2_ Capping	27.00	60.48	31.42	526.22	73.59	12.17	19.65	18.96
SC/SiO_2_/Ag-NPs (Ag: 5 nm)/SiO_2_ Capping	26.37	61.72	31.73	527.31	73.64	12.32	20.83	20.43
SC/SiO_2_/Ag-NPs (Ag: 7 nm)/SiO_2_ Capping	22.84	64.07	31.96	528.58	73.30	12.38	21.71	21.02

**Table 2 nanomaterials-11-01013-t002:** Average weighted reflectance (*R_W_*), average weighted EQE (*EQE_W_*), and photovoltaic performance of all evaluated solar cells.

Sample	R_W_ (%)	EQE_W_ (%)	J_SC_ (mA/cm^2^)	V_OC_ (mV)	FF (%)	η (%)	ΔJ_SC_ (%)	Δη (%)
	@350~1000 nm							
Bare SC (SC)	38.69	50.66	26.26	517.66	75.27	10.23	--	
SC/SiO_2_/SOG	29.59	57.96	31.35	524.92	73.15	12.04	19.38	17.69
SC/SiO_2_/SOG: Species-500	22.67	59.62	32.22	523.11	74.04	12.48	22.70	22.09
SC/SiO_2_/SOG: Species-550	22.66	60.10	32.25	523.32	74.18	12.52	22.81	22.39
SC/SiO_2_/SOG: Species-600	21.60	61.52	32.32	523.35	74.15	12.54	23.08	22.58

**Table 3 nanomaterials-11-01013-t003:** Average weighted reflectance (*R**_W_*), average weighted EQE (*EQE**_W_*), and photovoltaic performance of all evaluated solar cells.

Sample	R_W_ (%)	EQE_W_ (%)	J_SC_ (mA/cm^2^)	V_OC_ (mV)	FF (%)	η (%)	ΔJ_SC_ (%)	Δη (%)
	@350~1000 nm							
Bare SC	38.69	50.66	26.26	517.66	75.27	10.23	--	
SC/SiO_2_/SOG	29.59	57.96	31.35	524.92	73.15	12.04	19.38	17.69
SC/SiO_2_/SOG: Species-500	22.67	59.62	32.22	523.11	74.04	12.48	22.70	22.09
SC/SiO_2_/Ag-NPs (3 nm)/SOG: Species-500	19.83	62.05	32.91	527.52	74.26	12.89	25.33	26.02
SC/SiO_2_/Ag-NPs (5 nm)/SOG: Species-500	15.13	60.65	32.53	528.48	74.41	12.79	23.90	25.06
SC/SiO_2_/Ag-NPs (7 nm)/SOG: Species-500	14.45	59.74	31.99	527.52	73.69	12.44	21.83	21.56

## Data Availability

Not applicable.

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
