# Peer review of "Characterization of Plasmonic Scattering, Luminescent Down-Shifting, and Metal-Enhanced Fluorescence and Applications on Silicon Solar Cells"

_nanomaterials, 2021, doi:10.3390/nano11041013_

Round 1
Reviewer 1 Report
Ho and colleagues studied the effect of different experimental parameters on plasmonic properties of silver NPs for further applications on solar cells. The step-by-step properties are characterized in order to finally analyze the solar cell configuration combining all components.
I have only a few minor comments about the text:
- In figure 12, the absolute emission intensities are reported. Is there any possibility that the emission differences are due to slightly different thickness or concentration of emission species?
- In general, are the tests repeated few times on different samples prepared with the same experimental procedure? Sometimes the analyzed parameter values are not strongly different even if the precision on the reported values is very high. If the tests have been repeated, it should be more useful to report also the error for each measurement.
- Figure 14-b are corrupted.
- Some suggestions on English language: on line 11, I would suggest changing the verb “characterized” with “studied”; on line 13: “by combining the two processes to enhance”; on line 43: “to expand the usable range”; on line 61: “and their combination in terms of absorbance”.
I would suggest the publication of the manuscript in “Nanomaterials” journal after minor revisions.
Author Response
Comments and Suggestions for Authors
Ho and colleagues studied the effect of different experimental parameters on plasmonic properties of silver NPs for further applications on solar cells. The step-by-step properties are characterized in order to finally analyze the solar cell configuration combining all components.
Ans:
The authors would like to express their sincere appreciation to the reviewer for useful comments. The reviewers’ comments are taken into account and mentioned issues have been addressed for significant improvement in the revised manuscript. In keeping with the comments of the Reviewer #1, we have made some revisions by marking with the red color and underline at exact locations on the revised manuscript.
I have only a few minor comments about the text:
In figure 12, the absolute emission intensities are reported. Is there any possibility that the emission differences are due to slightly different thickness or concentration of emission species? In general, are the tests repeated few times on different samples prepared with the same experimental procedure? Sometimes the analyzed parameter values are not strongly different even if the precision on the reported values is very high. If the tests have been repeated, it should be more useful to report also the error for each measurement.
Ans:
- Thanks a lot for your
- The samples were prepared with the same experimental procedure and at 3 wt% concentrations.
- PL data obtained in this work, as shown in Figure 12, were measured 3 points on the sample and 3 times for each point to confirmed the intensity and wavelength of peak emission at room temperature. The emission intensity of species-500 was higher than that species-550 and species-600.
Figure 14-b are corrupted.
Ans:
- The better quality Figure14-b has been provided on the revised manuscript.
Some suggestions on English language: on line 11, I would suggest changing the verb “characterized” with “studied”; on line 13: “by combining the two processes to enhance”; on line 43: “to expand the usable range”; on line 61: “and their combination in terms of absorbance”.
Ans:
- Thanks for your suggestion and remarked.
- We revised as “This paper studied the plasmonic effects of silver nanoparticles (Ag-NPs), the luminescent down-shifting of Eu-doped phosphor particles, and the metal-enhanced fluorescence (MEF) achieved by combining the two processes to enhance the conversion efficiency of silicon solar cells”, on the revised manuscript.
- We revised as “This study characterized the plasmonic effects of silver nanoparticles (Ag-NPs), the luminescent down-shifting of Eu-doped phosphor particles, and the metal-enhanced fluorescence (MEF) achieved by combining the two processes to enhance the conversion efficiency of silicon solar cells”, on the revised manuscript.
- We revised as “One approach to expand the usable range of light involves the application of a conversion layer, such as down conversion (DC) [18,19]”, on the revised manuscript.
- We revised as “This paper systematically characterizes the effects of Ag-NPs, Eu-doped phosphor particles, and their combination in terms of absorbance, photoluminescence (PL), and optical reflectance spectra”, on the revised manuscript.
I would suggest the publication of the manuscript in “Nanomaterials” journal after minor revisions.
Ans:
The authors would like to express their sincere appreciation to the Reviewer for useful comments.

Reviewer 2 Report
In this manuscript, the authors suggested the characterization of plasmonic scattering, luminescent 2 down-shifting, and metal-enhanced fluorescence and applications on silicon solar cells. The authors explained that the plasmonic effects of silver nanoparticles (Ag-NPs), the luminescent down-shifting of Eu-doped phosphor particles, and the metal-enhanced fluorescence (MEF) achieved by combining the two to enhance the conversion efficiency of silicon solar cells.
The performance of device is impressive for SSCs research. And background analysis data is reasonable for understanding phenomenon.
I recommend this manuscript would be acceptable for Nanomaterials.
Author Response
Comments and Suggestions for Authors
In this manuscript, the authors suggested the characterization of plasmonic scattering, luminescent down-shifting, and metal-enhanced fluorescence and applications on silicon solar cells. The authors explained that the plasmonic effects of silver nanoparticles (Ag-NPs), the luminescent down-shifting of Eu-doped phosphor particles, and the metal-enhanced fluorescence (MEF) achieved by combining the two to enhance the conversion efficiency of silicon solar cells.
The performance of device is impressive for SSCs research. And background analysis data is reasonable for understanding phenomenon.
I recommend this manuscript would be acceptable for Nanomaterials.
Ans:
The authors would like to express their sincere appreciation to the Reviewer for useful comments.
Reviewer 3 Report
The authors systematically discuss the properties of various spectral conversion layers and their influence on the solar cell efficiency. The down conversion method is obviously known for a while already, but I left the field of solar cells many years ago, and thus it was unknown to me. Reading the paper, however, I immediately liked this method for it's elegance and simplicity.
The paper is definitely of good technical quality and also the scientific quality is sound. Thus, I can recommend it's publlication in general as it is. However, the authors might want to address briefly the following questions for the non expert readers:
- How large is the improvement in efficiency compared to other recently oublished conversion methods ?
- Are there other advantages besides the enhanced efficiency, like a better stability in harsh environments e.g.?
- Are the used materials a cost factor if the cells are used commercially ?
- Would a commercial use be technologically easy, or are there technological problems ?
Addressing these questions would also help to judge / point out the importance pf this work.
Author Response
Comments and Suggestions for Authors
The authors systematically discuss the properties of various spectral conversion layers and their influence on the solar cell efficiency. The down conversion method is obviously known for a while already, but I left the field of solar cells many years ago, and thus it was unknown to me. Reading the paper, however, I immediately liked this method for it's elegance and simplicity.
The paper is definitely of good technical quality and also the scientific quality is sound. Thus, I can recommend it's publication in general as it is. However, the authors might want to address briefly the following questions for the non-expert readers:
How large is the improvement in efficiency compared to other recently published conversion methods?
Ans:
In this study, a 26.02% increase in conversion efficiency (from 10.23% to 12.89%) was obtained due to MEF effects, which was higher than that the silicon cells coated with only a Ag-NPs layer or only a Eu-doped phosphor layer.
Are there other advantages besides the enhanced efficiency, like a better stability in harsh environments e.g.?
Ans:
- An additional enhancement of conversion efficiency of silicon solar cells can be achieved using the proposed approach, when the completed silicon cells through the post-process of coating a MEF layer.
- It is a good idea to study that the stability of the silicon solar cells enhanced by MEF effects under harsh environments. We can study this issues in the further.
Are the used materials a cost factor if the cells are used commercially? Would a commercial use be technologically easy, or are there technological problems?
Ans:
- In this work, the plasmonic scattering, luminescent down-shifting (LDS), and metal-enhanced fluorescence effects to enhance efficiency were demonstrated on the planar surface silicon solar cells using e-beam evaporation and spin-on film coating. Particularly, the average particle diameter of the used LDS materials was about 15 micrometers.
- In general, the commercial silicon solar cell was with textured surface, which presents a random arrangement of pyramidal structures of approximately 4-7 micrometers. Larger LDS-phosphor particles (e.g. > 10 micrometers) situated near the top of the pyramids tends to interfere with the anti-reflective properties of the texturing. The particles also shade the pyramidal structures beneath, thereby altering their anti-reflective and multi-reflective functionality.
- Therefore, nano-scale or sub-micro size LDS-phosphor particles are preferred for the commercial silicon solar cells with textured surface. In addition, the LDS-phosphor deposition methods for the commercial silicon solar cells with textured surface are still need to improve in the further.
Addressing these questions would also help to judge / point out the importance pf this work.
Ans:
The authors would like to express their sincere appreciation to the Reviewer for useful comments.
Reviewer 4 Report
In this manuscript, the authors investigated the plasmonic effects of Ag-NPs, LDS effects of Eu-doped phosphor particles, and the MEF effects of both in PL spectra, EQE, and solar cell performance. This work will be of special interest to the photovoltaic and photophysics community. However, there are some concerns that should be addressed before publishing the work in Nanomaterials.
- The title of the paper is not understandable for the general readership. I suggest: Effects of silver nanoparticles and Eu-doped phosphor particles on the efficiency of silicon solar cells.
- Line 171; Refractive index of air is 1 not zero. Please correct it.
- Please provide more detail about responsivity in the main text for figure 7.
Author Response
Comments and Suggestions for Authors
In this manuscript, the authors investigated the plasmonic effects of Ag-NPs, LDS effects of Eu-doped phosphor particles, and the MEF effects of both in PL spectra, EQE, and solar cell performance. This work will be of special interest to the photovoltaic and photophysics community. However, there are some concerns that should be addressed before publishing the work in Nanomaterials.
Ans:
The authors would like to express their sincere appreciation to the reviewer for useful comments. The reviewers’ comments are taken into account and mentioned issues have been addressed for significant improvement in the revised manuscript. In keeping with the comments of the Reviewer #4, we have made some revisions by marking with the blue color and underline at exact locations on the revised manuscript.
- The title of the paper is not understandable for the general readership. I suggest: Effects of silver nanoparticles and Eu-doped phosphor particles on the efficiency of silicon solar cells.
Ans:
(1) Thanks a lot for your suggestion that the title of the paper for the general readership.
(2) This paper studied the plasmonic effects of silver nanoparticles, the luminescent down-shifting of Eu-doped phosphor particles, and the metal-enhanced fluorescence (MEF) achieved by combining the two to enhance the conversion efficiency of silicon solar cells. Because the MEF effects to enhance the conversion efficiency of silicon solar cells is one of the point of the study, the title of the paper is maintained the previous title.
- Line 171; Refractive index of air is 1 not zero. Please correct it.
Ans:
Thanks a lot for your remarked.
We revised as “The absorption range and intensity were further enhanced by the SiO2 capping layer, which altered the refractive index of the medium surrounding the Ag-NPs from 1.0 (air) to 1.45 (SiO2)”, on the revised manuscript.
- Please provide more detail about responsivity in the main text for figure 7.
Ans:
Thanks a lot for your remarked.
We add that “The responsivity of the silicon p-n diodes was approximately 0.15, 0.20, and 0.26 A/W at the peak emission wavelength of species-500, species-550, and species-600, respectively”, on the revised manuscript.
